# Identification of Key Regulatory Pathways of Basidiocarp Formation in *Pleurotus* spp. Using Modeling, Simulation and System Biology Studies

**DOI:** 10.3390/jof8101073

**Published:** 2022-10-13

**Authors:** Anupam Barh, Kanika Sharma, Pankaj Bhatt, Sudheer Kumar Annepu, Manoj Nath, Mahantesh Shirur, Babita Kumari, Kirti Kaundal, Shwet Kamal, Ved Parkash Sharma, Sachin Gupta, Annu Sharma, Moni Gupta, Upma Dutta

**Affiliations:** 1ICAR-Directorate of Mushroom Research, Solan 173 213, India; 2Department of Agricultural & Biological Engineering, Purdue University, West Lafayette, IN 47906, USA; 3ICAR-Indian Institute of Soil and Water Conservation, Research Center, Udhagamandalam 643 006, India; 4National Institute of Agricultural Extension Management (MANAGE), Hyderabad 500 030, India; 5Division of Plant Pathology, Faculty of Agriculture, Sher-e-Kashmir University of Agricultural Sciences & Technology of Jammu, Jammu 180 009, India; 6Department of Plant Pathology, College of Horticulture, Dr. Yashwant Singh Parmar University of Horticulture and Forestry, Nauni, Solan 173 230, India; 7Division of Biochemistry, Faculty of Agriculture, Sher-e-Kashmir University of Agricultural Sciences & Tech-nology of Jammu, Jammu 180 009, India; 8Division of Microbiology, Faculty of Agriculture, Sher-e-Kashmir University of Agricultural Sciences & Tech-nology of Jammu, Jammu 180 009, India

**Keywords:** oyster mushroom, pinhead formation, SBML, network analysis, Cell Designer

## Abstract

*Pleurotus* (Oyster mushroom) is an important cultivated edible mushroom across the world. It has several therapeutic effects as it contains various useful bio-molecules. The cultivation and crop management of these basidiomycete fungi depends on many extrinsic and intrinsic factors such as substrate composition, growing environment, enzymatic properties, and the genetic makeup, etc. Moreover, for efficient crop production, a comprehensive understanding of the fundamental properties *viz*. intrinsic–extrinsic factors and genotype-environment interaction analysis is required. The present study explores the basidiocarp formation biology in *Pleurotus* mushroom using an in silico response to the environmental factors and involvement of the major regulatory genes. The predictive model developed in this study indicates involvement of the key regulatory pathways in the pinhead to fruit body development process. Notably, the major regulatory pathways involved in the conversion of mycelium aggregation to pinhead formation and White Collar protein (*PoWC1*) binding flavin-chromophore (FAD) to activate respiratory enzymes. Overall, cell differentiation and higher expression of respiratory enzymes are the two important steps for basidiocarp formation. *PoWC1* and *pofst* genes were participate in the structural changes process. Besides this, the *PoWC1* gene is also involved in the respiratory requirement, while the *OLYA6* gene is the triggering point of fruiting. The findings of the present study could be utilized to understand the detailed mechanism associated with the basidiocarp formation and to cultivate mushrooms at a sustainable level.

## 1. Introduction

*Pleurotus* is a widely grown edible mushroom and its rank moved from second to first position. *Pleurotus* contributes to 26% of global mushroom production [1,2]. The huge species diversity of this genus, as well as its ability to utilize the different types of agricultural wastes or substrates, makes its adaptation favorable in different climates. This mushroom is nutritionally rich with varied medicinal benefits. For instance, many bioactive compounds are reported in oyster mushrooms such as polysaccharides, proteins, lectins, phenolic compounds, terpenoids and other useful myco-compounds [3]. Moreover, its other usages, such as bioremediation [4], dye making [5], cosmetics [6], etc., have already been reported in the last decade. For the successful production and effective crop handling of *Pleurotus*, knowledge of fruiting is essential, but the complete fruiting mechanism has not yet been fully understood. The biological phenomenon of transition from vegetative phase to reproductive phase in mushrooms is always a topic of curiosity amongst researchers. Since mushrooms contribute to a healthy diet vis à vis being storehouses of bioactive compounds, it is important to understand the mechanisms of fruiting to cultivate mushrooms at a sustainable level. *Pleurotus* production is largely concentrated in Asian countries such as China, Japan, South Korea, Taiwan, Thailand, Vietnam and India. Besides the Asian countries, it is significantly grown in more than 25 countries. The *Pleurotus* species is the largest among cultivated edible mushrooms at global level. There has been a rapid increase in *Pleurotus* cultivation in the last three decades. The total production registered a many-fold increase from 1987 to 2019. In 2019, the total global production of *Pleurotus* was recorded at around 11.18 million tonnes [2]. The life cycle of *Pleurotus* begins as spores, which germinate and form mycelial threads that colonize on lignocellulose wastes. The mycelia digest the substrate via its active enzyme system and convert it into colonized mycelial mass. The colonized substrate produces the fruit bodies under favorable conditions. The fruit body produces a white-colored spore mass from the gills. Compatible spores with different mating types fuse to produce a dikaryotic mycelium which continues the life cycle in nature. This life cycle in *Pleurotus* is known as the heterothallic life cycle [7]. *Pleurotus* as a genus is well known for its species diversity and easy cultivation practices. However, its full productivity is yet to be realized due to the limited availability of potential germplasm with variable response to the growing environment and the agronomic practices. The genotype × environment interaction is a critical factor to enhance the *Pleurotus* productivity, as environmental factors are directly linked to its fruiting initiation, development and maturity. The fruiting of oyster mushrooms is affected by light, humidity, pH of the substrate, oxygen concentration, and temperature, and thus plays a crucial role in the fruiting mechanism [8]. For indoor cultivation, temperature and other parameters are variable for different *Pleurotus* species [9]. The variation in these parameters may cause drastic changes in the yield. Linking the external stimulus of fruiting to the molecular mechanism is essential for a uniform production cycle. The genotype×environment interactions can be better explained when all the intrinsic and extrinsic factors are correlated in a systematic approach. There are several metabolic and molecular pathways that regulate the fruiting development in *Pleurotus* [10]. In response to the interaction of their intrinsic (genetic) and extrinsic (physical, chemicals) factors, the cells will undergo a differentiation process within the fruit body’s developing tissues [11]. All these phenomena lead to the creation of the pattern that creates the specific distribution of differentiated tissues in the multicellular structure. The initiation and development of the fruit body in *Pleurotus* requires different stages *viz*. hyphal knot formation, primordial initiation, primordial formation and fruit body formation [10,12]. Various studies have been conducted in other mushrooms to understand the molecular mechanism of fruiting. In model fungal species, the studies on fruiting mechanism were conducted in *Schizophyllum commune* [13], *Coprinopsis cinerea* [14] and *Agrocybe aegerita* [15]. In most of the basidiomycetes mushrooms, the whole process of fruit body development fungus deploys mechanisms for communication, cell differentiation and development [16]. Theoretically, after hyphal knot converts into primordia, these primordial structures convert into mature fruiting body under suitable environmental conditions. The lower part of primordia forms the stem/stipe while upper parts forms cap or pileus [17]. In many studies, certain compounds/proteins are found to be responsible for fruiting in mushrooms, such as ostreolysin/hemolysins, [18] hydrophobin [19], lectins [14], oxidative enzymes/cytochrome 450 [20], and expansins [21]. The complexity involved in the total fruiting process make it difficult to understand whole process in one frame. Bioinformatics can provide an opportunity to solve the complex puzzle of the mechanism even in paucity of information. Using an in silico study, we can perform more accurate analysis without expending the chemicals in a wet laboratory [22]. Therefore, based on the available literature, we have linked the system biology approaches to construct the predictive model for the fruiting mechanism in the *Pleurotus* mushroom. The studies were also undertaken on simulation, rate kinetics of constructed model, network analysis and topological studies of network of fruiting in *Pleurotus* spp.

## 2. Materials and Methods

### 2.1. Development of Model

To find out the molecular mechanism of fruit body development with response to external stimulus in the *Pleurotus* mushroom, the literature was accessed by various sources, such as Pubmed, Google Scholar, and Research Gate, etc. The role of the genes responsible for fruiting and their probable mechanisms were noted and the system biology was studied. Using systems biology graphical notation (SBGN), the system biology studies were conducted in Cell Designer4.1. The Cell Designer software enables the biologist to depict biological process (catalysis, inhibition, activation, phosphorylation, etc.) and molecules (DNA, RNA, proteins, etc.) in cells as a diagram/model to show the biological network pathways using graphical notation symbols. In the present study, a model was constructed consisting of a biological network (Figure 1) and pathways to understand the biology of fruit body development in *Pleurotus* spp. Complex Pathway Simulator (COPASI), Systems Biology Markup Language (SBML) and ordinary differential equation (ODE) Solver were used for channels simulations in Cell Designer software for identification of principal parameters, variable quantity and simulation. Cell Designer required the Java programming language for operations on operating system (OS) platforms of Windows [23]. The simulation work was processed on a hardware configuration comprised of HP Pavilion Intel core-i5-7400T CPU with 2.40 GHz Processor and 8 GB RAM on 64 bit Windows OS with Java compatible for both 32 and 64 bit environment.

### 2.2. Kinetic Rate Reaction Calculation and Model Simulation

In the constructed model, each reaction was calculated for kinetic rate equations using SBML (Systems Biology Markup Language) squeezer version plugin 1.3 of Cell Designer. SBML is a machine-readable format that helps in describing biological processes such as metabolic pathways, cell signaling pathways, etc. [24]. The rate equations generated using Cell Designer help in terms of being error-free and quick compared to manual calculation. Moreover, a large set of V_max_ (maximal reaction rate) calculations for each reaction can be performed for a complicated model. The SBML squeezer plugin has a unique feature which harvests data from the model of all biological components in SBGN representation and then chooses to withdraw the necessary data from the Systems Biology Ontology (SBO) annotations [25,26]. The simulation and dynamics of constructed model were studied using SBML ODE Solver Library (SOSlib) in Cell Designer. SOSlib is a programming library used for study of chemical reaction network models utilizing symbolic and numerical analysis encoded in the SBML. The SOSlib in applied computing is used for computational biology in genetics studies, system biology, etc. [27]. Quantitative investigation of biological networks was performed by normal differential equations and ODE-based simulations through graphical user interface.

### 2.3. Network Analysis

Network Analyzer is an effective and efficient tool present in Cytoscape that enables us to compute topological network parameters in both undirected and directed networks [28]. This open-source software helps in integrating, examining, and visualizing any biological and complex networks. Cytoscape has the ability to add on third-party-developed tools to improve visualization and analysis [29]. The Biological Network Manager (BiNoM) plugin (version 2.5) of Cytoscape was used. The plugin is useful to import Cell Designer SBML formats files [30]. Interactive networks representation with different styles can be brilliantly achieved by Cytoscape [31]. Using Cytoscape ver. 2.8.3, *Pleurotus* spp. fruiting mechanism model was visualized for molecular interaction. The data of the model developed by Cell Designer was exported and used in Cytoscape. BiNoM is also used for analyzing complex biological networks using SBML, SBGN, and BioPAX formats for complicated network structures. In the present study, we used Cytoscape for investigating and visualizing the crucial genes, proteins and paths responsible for *Pleurotus* fruiting.

## 3. Results

For *Pleurotus* spp., previous studies were conducted on various extrinsic and intrinsic factors that affect fruiting but no clear mechanisms or pathways have been suggested to understand the biology of fruit body development. Using system biology graphical notation (SBGN), we tried to establish a fruit body development model accommodating some of the most important environmental parameters responsible for fruiting. Here, the models were created to attempt to find out the responsible key genes, enzymes and pathways to obtain insight into the fruit body development as well as the effect of the environment on the development and biology. The models showed similarities to earlier studies found in the existing literature and identify some key components responsible for fruiting in *Pleurotus* [17]. The model comprises four compartments (environment, fungal cell, mitochondria and substrate), 53 species, 11 genes, 11 mRNA, 16 proteins and 39 reactions (Figure 2).

### 3.1. Dynamic Behavior Studies and Model Simulations

The prediction and behavior by simulation are used to understand the series of events taking place during the fruit development process in mushrooms. The predicted dynamic behavior ascertains the effect of the expression of genes in different environments. Simulation studies are helpful in the interpretation of the fate of various molecular species and metabolites with the degree of importance in the model. In simulation, although it is difficult to estimate the amount or concentration of molecular species/metabolites, an approximation was set up using previous wet laboratory-based studies. Knowing the amount of molecular species in quantitative terms will help us to find links and association amongst species [25,32]. In our model, we have used the range of 0.5 to 2.5 for each molecular species. The values of genes were set at 0.5, proteins were set at 2.0 and mRNA was set at 1.0. The value 2.5 was set for phenotypes and phenotypic responses. In each species, the variation of 0.5 to 1.0 was given and checks the changes in expression in the fungal cell (Figure 3, Figure 4 and Figure 5). Simple molecules were set at 0.5 due to its basal amount. The values for Pleurotolysin and membrane receptors were between 0.0 and 1.0 to study the effect on mycelium aggregation or knot formation. Similarly, the values for hydrophobin and *Pleurotus* White Collar complex protein were between 0.0 and 2.0 to study the effect on pinhead formation and fruit body devolvement. All the dynamics are provided in Appendix A.

### 3.2. Topological Analysis of Fruit Body Development 

The model analyzed for the network analysis showed the series of events predicted to occur during fruiting, and a total of 89 nodes and 90 edges were developed. The simple topology parameters are number of nodes, number of edges, connected components, clustering coefficient, network diameter, network radius, shortest paths, average neighbors numbers, density, radius, heterogeneity, centralization, self-loops and characteristic path length [28]. The network was developed using a directed network to estimate the average path length. The simple parameters analyzed in the network analysis are presented in Table 1.

The clustering coefficient of a node is the degree to which the nodes cluster together. The value range is between 0 to 1. The clustering coefficient is the ratio of the number of edges between the neighbors to the maximum number of edges that could possibly exist between the neighbors [33]. The number of connected components indicates the connectivity of a network and is inversely related. The neighborhood connectivity distribution shows that there is a decreasing function in *k*, which means that in the network, the edges are between low-connected and highly connected nodes [34] (Figure 6). The network diagram shows the node size “in-degree” and node color “betweeness centrality” and can be visualized in Figure 6. The map helped in the determination of the hub nodes. The nodes represented in red color are hub nodes and have a significant regulatory character in *Pleurotus* fruiting. The other nodes in yellow color had the least regulatory effect on the fruiting mechanism pathways. The green-color nodes show the moderate effect on the model pathway (Figure 6). 

In the model, the fruit body development process involves most of the regulatory pathway re-16. The other important regulatory pathways were the conversion of mycelium aggregation to pinhead formation re-6 and White Collar protein (*PoWC1*) *PoWC1*-binding flavin-chromophore FAD to activate respiratory enzymes re-37. The other important pathways according to importance were re-20, re-24, re-27, re-12, re-14, re-22, re-3, re-36, re-25, re-39, re-4 and re-9 (Figure 6).

In network analysis, the edges and nodes are connected in terms of degree. In the directed node approach, in-degree distribution estimates the edges approaching a node, whereas the out degree distribution takes in account those edges which are leaving the node [35]. The nodes can be treated differently for different networks, for example in metabolic network, reactions are taken as edges and the metabolic intermediates are observed as nodes, while in gene regulatory networks, the nodes are genes and its regulatory molecules are edges. Moreover, in the protein–protein interactions network, nodes are proteins and edges are bonds between them [36]. The biological properties of the hubs are important in the fruiting mechanism to judge the role of protein or genes. Hubs are usually an intermediate, gene or protein with key regulatory roles (Figure 7). In the model, some highly connected nodes, such as *Pleurotus* White Collar protein (*PoWC1*) protein, are important in fruiting and mostly in the fruit body development. The *Pleurotus* mushroom is considered as one of the most important mushrooms not only for its edible use but also for its myco-remediation potential [37]. It is also essential that its fruiting be propagated in the environment to liberate the spores. The fruiting mechanism still needs to be studied in detail. A study showed that 596 unigenes were expressed during fruit body formation [38]. In, *Pleurotus* genetic improvement this research may be helpful in the development of early varieties using the important fruiting genes (Table 2).

## 4. Discussion

To find more detailed fruiting mechanism pathways and their response to different environmental factors, an in silico experiment was designed. Under the specific environmental conditions, development of the fruiting body starts with mycelium/aerial hyphal aggregation. Nutrient, light, humidity and temperature, etc., are some of the factors responsible for fruit body development [43,50,51]. These aerial dikaryotic hyphal aggregates continuously develop into primordia/pinhead which further differentiate into the mature mushroom fruiting body [51]. In our study, we broadened our knowledge on how different environmental factors, developmental genes (receptors genes, expression genes) and enzymes are responsible for the molecular mechanisms of fruiting in *Pleurotus* [52]. The predictive model helps to gain an insight into various genes and enzymes and their temporal expression which affects the fruiting mechanisms. Through studying *Pleurotus*, it is known that Pleurotolysin has 138 amino acid residues (16 k Da) and a pore-forming protein has been found to have a role in fruiting [39]. The *OLYA6* gene, also known as Ostreolysin A6 (Aa-pri1 gene) transcript, is found in primordial stages and in immature fruit bodies. These proteins are associated with the membrane-attack complex/perforin (MACPF) domains and have various biological roles, such as defense and attack, organism development, cell adhesion and signaling [53]. Protein Pleurotolysin consists of non-associated components A and B. Pleurotolysin A is an egerolysin-like protein, while Pleurotolysin B has a MACPF domain [53]. It is observed that Pleurotolysin A and B together can cause hemolytic activity. In some of the studies, it is seen that Pleurotolysin A interacts with the membrane by recognizing the polar head group of sphingomyelin and eventually interacts with the hydrophobic ends of cholesterol and other phospholipids [54]. Moreover, the presence of both cholesterol and sphingomyelin affects Pleurotolysin binding to mono- and bilayer lipids, and lipid vesicle permeabilization. There are various explanations for the interaction, including that the Lysophospholipids act as a signaling molecule and thus regulate the pore forming ability of Pleurotolysin which transforms the cell differentiation from a vegetative to reproductive phase [55]. It is also noted in various studies that Pleurotolysin cannot increase the permeability of the lipid vesicle [56], but they may bind to it.

During the simulation of the network, we have found that reaction-3 (re3) (Figure 6) had some role in cell differentiation but we could not find a major role in the whole basidiocarp development. Although, it can be assumed that *OLYA6* may be the triggering point of fruiting. Phenotypically hyphae aggregation may be governed by this protein. In our simulation study, it was found that the Pleurotolysin is an important protein for mycelium aggregation (Figure 3).

Another important step after mycelium aggregation is pinhead formation. In a humid environment, the hydrophobin gene is the most highly expressed gene of all during fruit body development. Hydrophobin is cysteine-rich and abundantly expresses fungal proteins [57,58]. Hydrophobins lower surface tension, letting the hyphae escape and initiating pinhead growth (aerial structure, aerial hyphae and fruit body) studied in *Schizophyllum commune* [40,41]. In the model, the hydrophobin *Fbh1* gene is transcribed and translated in high humidity and produces hydrophobin proteins that interfere in aerial hyphae, spores and fruiting body development. Structurally, the Fbh1 and POH1 hydrophobin proteins contain two cysteine and 113 amino acids clusters [59]. During basiodiocarp formation, the hydrophobin protein is involved in hyphal adhesion and protects against desiccation and helps in gas exchange regulation. In *Pleurotus ostreatus*, the *POH1* (*P. ostreatus* hydrophobin 1) gene showed a regulating effect for limiting pinhead development while *POH2* and *POH3* expressed specific levels of hydrophobins at the vegetative phase [60]. Penase et al., in 2002 and 2004, investigated two genes, i.e., *vmh3* and *Fbh1* from *P. florida* that were expressed in vegetative mycelium and fruiting body development, respectively [42,43]. Various other hydrophobins are reported in other mushrooms such as in the bracket mushroom (SC1, SC3 and SC4) [61], while in the button mushroom, ABH1 and ABH2 were detected in the mature fruit body [41,62].

In simulation network, reaction no. 6 (re6) in Figure 6 is also a significant reaction catalyzed by hydrophobin and the Pofst protein. This reaction is more important than reaction 3 in terms of significance as represented by the network analysis diagram. The *Pofst3* gene is responsible for controlling the number and shape of sporocarps. *Pofst3* is involved in the regulation by inhibiting the cluster formation in *P. ostreatus* [44]. Hyphae aggregation may be governed by this protein. In our simulation study, it was seen that the Pleurotolysin is an important protein for mycelium aggregation. It is seen that hydrophobin and the Pofst protein play an important role in pinhead formation. In reaction dynamics, as the concentration of hydrophobin increases, the pinhead number also increases, but a high concentration of the Pofst protein decreases the number of pinheads, but in the presence of both proteins a regulated amount of pinhead is formed (Figure 4). Pinhead to fruit body formation is a more complex process governed by various genes which include structural genes, metabolic genes, respiratory genes and secondary metabolites genes, etc. In *Pleurotus,* as reported in studies, blue light significantly contributes to the growth of pileus [49]. Transcriptomics study showed that blue light enzymes, mainly glyceraldehyde-3-phosphate dehydrogenase (GAPDH), 6-phosphogluconate dehydrogenase (6PGD) and phosphoenol pyruvate carboxykinase (PEPCK), significantly upregulated pileus formation by using specific pathways such as glycolysis/gluconeogenesis and pentose phosphate pathway (PPP) where 6PGD and PEPCK are rate-controlling genes in the pentose phosphate pathway and gluconeogenesis, respectively, and GAPDH is crucial in the glycolysis pathway [45]. A photoreceptor phototropic-like protein called *Pleurotus* White Collar 1 combines with FAD (flavin adenine dinucleotide) to form the *PoWC1*–FAD complex that induces the transcription and expression of respiratory genes by binding to their promoter site and inducing gene expression that leads to fruiting body development. In the model, it can be seen that both Glucose-6 phosphate dehydrogenase (G6PD) and phosphofructokinase (PFK) enzymes are involved in phosphoenol pyruvate (PEP) formation (Figure 2). The G6PD enzyme is mainly representative of the PEP pathway, but it showed lower activity as compared to hexokinase and glucose-6 phosphate isomerase (glycolytic pathway enzymes) for mycelial and non-sporulating stages. At the time of sporulation, the glycolytic pathway enzymes increase significantly as compared to other mycelial stages in the *P. ostreatus* mushroom [63]. A similar study, by Schwalb in 1974, reported that, in *Schizophyllum commune*, the sugar metabolism, glycolytic pathway was a major route during fruit body development [64]. After PEP product, the 3-deoxy-D-arabinoheptulosonate 7-phosphate synthase (DAHP) enzyme participated, and the end result is the formation of shikimic acid, collectively it is known as a shikimic acid pathway [65]. PEP entered into the TCA (tricarboxylic acid cycle) cycle in mitochondria followed by the electron transport chain (ETC) reaction to generate ATP via oxidative phosphorylation. The mitochondrial ETC consists of different complexes and these complexes form reactive oxygen as a byproduct during ATP formation and any imbalance or stress, especially heat stress, or recovery in their antioxidant systems [66,67]. During this phase, blue light also helps in the expression of genes encoding carbohydrate-active enzymes (CAZymes) during primordial differentiation to the fruiting body [46]. Moreover, Laccase is also one factor that is responsible for fruit body structuring and secondary metabolite formation in the *Pleurotus* mushroom. Laccases are multi-copper polyphenol oxidase enzymes which help in delignification, pathogenesis and pigmentation of the fruiting body [68,69]. *PoLac1–12* putative laccase genes clusters have been noted in the genome of the *P. ostreatus* mushroom. As a matter of fact, during a transcription regulation study of Laccase genes, they were found to be involved in various stages, including mycelium, primodium and fruiting body formation [47]. The *PoLac 12* gene is supposed to be responsible for lignin degradation after the formation of the mRNA-Polac12 complex. This complex finally degrades lignin and helps to develop primordia and the fruiting body of *P. ostreatus* [70]. Other Laccase genes such as *PoLac2* and *PoLac* (*POXC*) have been overexpressed in the formation of the prophase of mycelium vegetative growth in *P. ostreatus* and are so closely related to lignin degradation [71]. In network analysis, it was found that reaction 13 and reaction 37 are the most significant. Both reactions are linked with fruit body development. In reaction 13, mostly PAL genes, Laccase and the ATP development process are important, while a significant role can be observed for the *PoWC1* protein (Figure 6). In simulation studies, it is seen that, as the *PoWC1* protein increases, the fruit body development increases. Moreover, PAL1, PAL2, G6PD, Laccase, structural proteins, and electron transport chain enzymes have positive and direct effects on the body development of *Pleurotus.*

Some other genes shown in the model, such as Phenylalanine ammonium-lysae (*PAL*), are protein genes, which were highly expressed in the phenylpropanoid pathway [72], that develop the phenolic compound in fruiting bodies. *PAL* genes expression are found under abiotic stresses, mainly UV light, temperature, salts, etc., that lead to the collection of phenolic compounds [47]. Two *PAL* genes, i.e., *PAL1* and *PAL2* are identified in *P. ostreatus* genome during the developmental stage, the expression of both genes was shown in all developmental stages but the highest expression was observed in fruit gills [73]. The expression of the *PAL2* gene significantly up-regulated the process during primordia, fruiting body and spore development in *P. ostreatus* [48]. *PALs*, along with tyrosinases, are involved in pileus pigment formation [49].

## 5. Conclusions

This study widens our knowledge of the advancements in the omics analysis technology by using different bioinformatics tools, demonstrating how the developmental and regulated genes/enzymes in the model are responsible for the fruiting body development of the edible *Pleurotus* mushroom. We studied a series of genes in our study and three genes showed importance based on the role of regulatory pathways, i.e., *PoWC1* (blue light), the *Fbh1* gene (hydrophobin) and the *Pofst* gene. *PoWC1* is involved in fulfilling both structural changes and helps to fulfill respiratory requirements, while the *Fbh1* gene (hydrophobin) and *Pofst* gene are mainly responsible for structural change. Moreover, *OLYA6* may be the triggering point of fruiting but does not act as the governor of key regulatory pathways. Several other biosynthesis genes (*PAL*, *PoLAC* gene) and respiratory enzymes (G6PD, PEP, DAHP synthase, PFK) are regulating the entire fruiting body development process during the primordial, fruiting body and spore development process in the *Pleurotus* mushroom. To understand the specific role of differentially expressed genes might help determine good candidates for in-depth experimental studies. Additional comprehension of different environmental factors and genes involved enhancing or supporting fruiting body induction and development is censorious from the trade point of view and will be helpful in the future for commercial *Pleurotus* mushroom cultivation.

## Figures and Tables

**Figure 1 jof-08-01073-f001:**
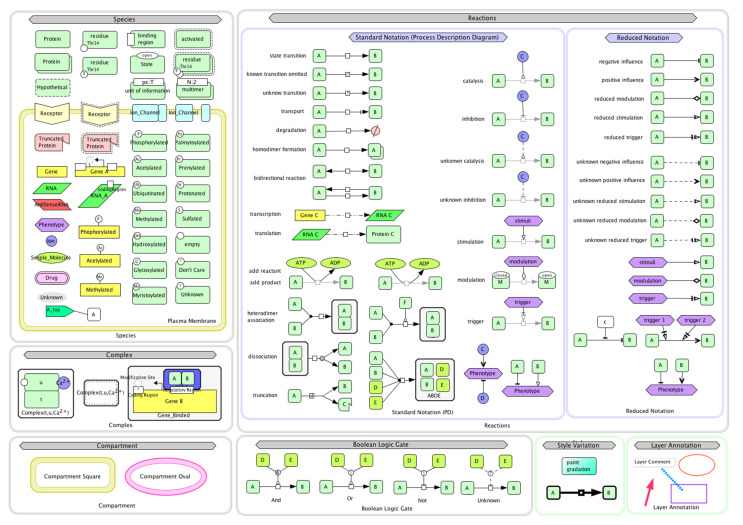
System biology graphical notation (SBGN) symbols used in Cell Designer for modeling of fruiting mechanism in *Pleurotus*.

**Figure 2 jof-08-01073-f002:**
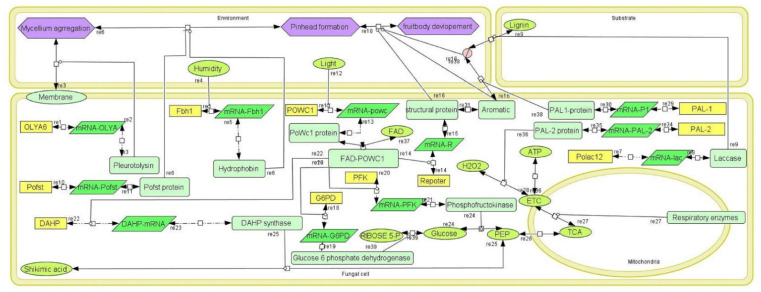
Fruiting mechanism in *Pleurotus* constructed using Cell Designer in fungal cells, in connection with the substrate and environment. The diagram shows the simple molecules, proteins, mRNA, DNA, enzymes, catalysis, and inhibition, etc.

**Figure 3 jof-08-01073-f003:**
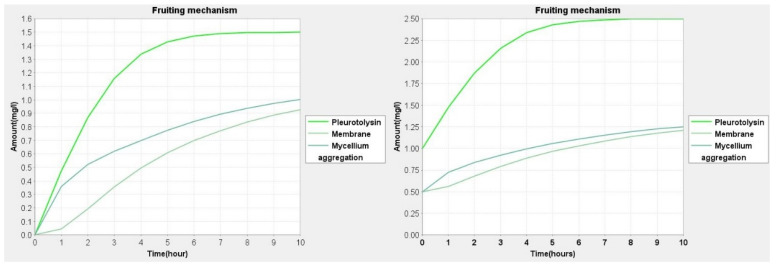
Simulation curve for Pleurotolysin protein and membrane receptor with respect to mycelium aggregation, the amount of Pleurotolysin was set at 0.0 and 1.0, respectively, while and membrane receptor and mycelium aggregation were set to 0 and 0.5.

**Figure 4 jof-08-01073-f004:**
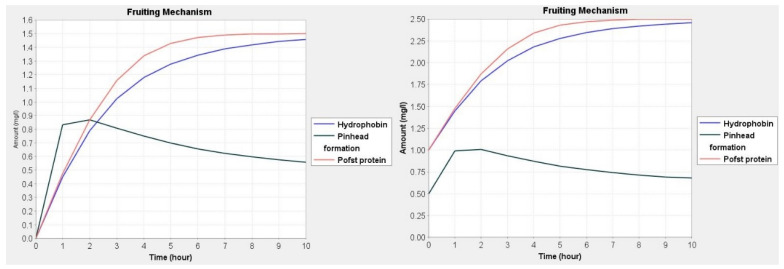
Dynamics of hydrophobin protein and Pofst protein with respect to pinhead formation, the amount of hydrophobin and pinhead formation were set at 0.0 and 1.0, respectively, while Pofst protein was set at 0.0 and 0.5.

**Figure 5 jof-08-01073-f005:**
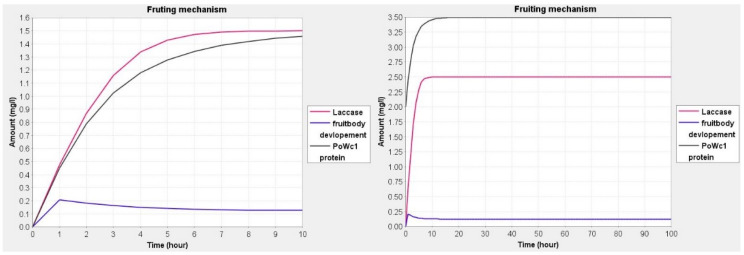
Dynamics of *PoWC1* protein and Laccase with respect to fruit body development, the amount of *PoWC1* protein was set at 0.0 and 2.0, respectively, while Laccase and fruit body development were set at 0.0.

**Figure 6 jof-08-01073-f006:**
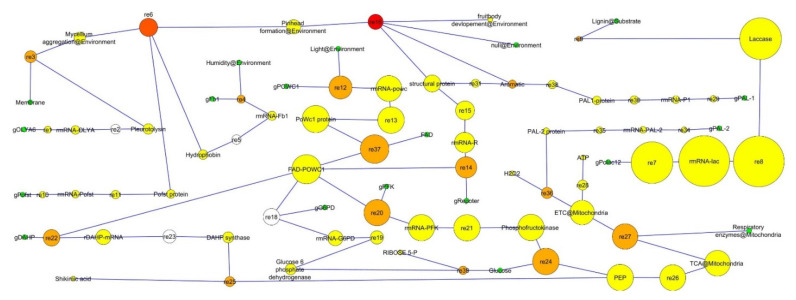
Mapping of the fruiting mechanism pathway: visualization of complete network pathway to map hub nodes.

**Figure 7 jof-08-01073-f007:**
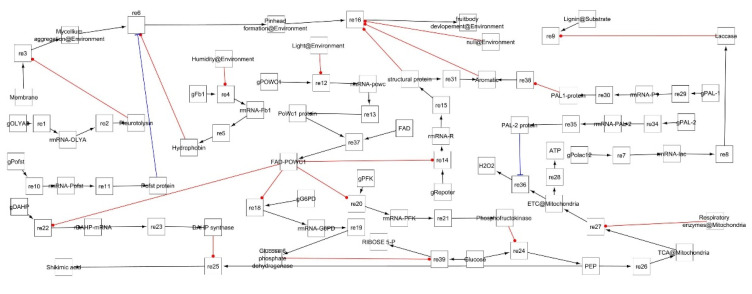
Module style view of fruiting mechanism pathway network node with physical simulation and catalysis.

**Table 1 jof-08-01073-t001:** Simple topology parameters of fruiting network.

Sl. No.	Simple Topology Parameters	Values
1.	Clustering coefficient	0.0
2.	Connected components	2.0
3.	Network diameter (largest distance between two nodes)	18.0
4.	Network radius	1
5.	Shortest paths	747(%)
6.	Characteristic path length/average shortest path length	5.63
7.	Average number of neighbors/average connectivity	2.02
8.	Network density	0.0

**Table 2 jof-08-01073-t002:** Role of genes and protein in *Pleurotus* fruit body formation and development.

Sl. No.	Protein	Role	Reference
1	Pleurotolysin	Pore-forming protein having role in fruiting	[39]
2	Hydrophobins	Initiate pinhead growth makes aerial structure-aerial hyphae and fruit body	[40,41,42,43]
3	Pofst protein	Regulation of fruiting by inhibiting the excessive cluster formation	[44]
4	Glyceraldehyde-3-phosphate dehydrogenase (GAPDH)	Role in glycolysis/gluconeogenesis and pentose phosphate pathway to promote fruitbody growth	[45]
5	6-phosphogluconate dehydrogenase (6PGD)	Role in glycolysis/gluconeogenesis and pentose phosphate pathway to promote fruitbody growth	[45]
6	Phosphoenol pyruvate carboxykinase (PEPCK)	Role in glycolysis/gluconeogenesis and pentose phosphate pathway to promote fruitbody growth	[45]
7	*Pleurotus* White Collar protein	It induces transcription and expression of respiratory genes	[46]
8	Carbohydrate-active enzymes	Primordial differentiation to fruiting body with help of blue light	[46]
9	Laccases	Role in fruit body formation	[47]
10	*PAL* genes	Expression upregulated during primodia fruiting body and spore development. These genes along with tyrosinases involved in pileus pigment formation	[48,49]

## Data Availability

Not applicable.

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
