# Peer review of "Identification of Key Regulatory Pathways of Basidiocarp Formation in Pleurotus spp. Using Modeling, Simulation and System Biology Studies"

_jof, 2022, doi:10.3390/jof8101073_

Round 1

Reviewer 1 Report

You did an excellent job, but I think that ref 6 and 7 should be from a more recent period

Author Response

  1. The manuscript is revised as per the reviewer comment. The English spell check done and Grammer corrected
  2. Relevant citation and information added
  3. 6 and 7 citation are updated 

Reviewer 2 Report

Dear authors,

The article was well designed and discussed the Pleurotus mushroom basidiocarps formation. However, the authors expressed several hypothetical views in the article. In addition, the figures quality was poor. If possible the authors should include the Table that contains “mechanisms of basidiocarps development with valid references [metabolites, proteins, enzymes, gene, signaling molecules etc.]. 

I need some clarifications in your Article entitled “Identification of key regulatory pathways of basidiocarps formation in Pleurotus spp. using modeling, simulation and system biology studies,” which are as follows

Abstract

Queries: Line 23: deleted the double space between “and involvement” and “genes. The”.

Line no 20: Moreover, a comprehensive understanding of the fundamental properties viz. intrinsic-extrinsic factors and genotype-environment interaction analysis is required for efficient crop production.

To rewrite– “Moreover, for efficient crop production, a comprehensive understanding of the fundamental properties viz. intrinsic-extrinsic factors and genotype-environment interaction analysis”.

Line 29 – 32: Rewrite the sentences better to understand and spell check [strucrtural - structural, Beside – Besides, fromation- formation].

Line 31: The findings of the present study could – rewrite as “The present study findings could be…..” levels change as “level” 

Introduction

Line 54: In 2013, the total global production of Pleurotus was recorded around 6610000 tonnes [7]. “Update the sentence and include the most recent production range” 

Line no 56: spores germinates change as “germinate”

Author Response

  1. Figure quality in paper improved
  2. Table of various gene, protein (table-2) added with reference
  3. Line no 23 double space removed
  4. Line no 20 changed as per reviewer
  5. Line no 29 to 32 changed
  6. Line no 31 changed
  7. Line no 54 reference updated
  8. Line no 56 changed
  9. spell check done in whole manuscript
  10. Grammar checked in whole manuscript
